# The impact of the COVID-19 epidemic on students' mental health: A cross-sectional study

**Nikola Mirilović[1], Janko Janković[2]\*, Milan Latas[3,4]**

**1** Zemun Gymnasium, Zemun, Belgrade, Serbia, **2** Institute of Social Medicine, Faculty of Medicine, University of Belgrade, Belgrade, Serbia, **3** Faculty of Medicine, University of Belgrade, Belgrade, Serbia, **4** Clinic for Psychiatry, University Clinical Center of Serbia, Belgrade, Serbia

\* janko.jankovic@med.bg.ac.rs

**Data Availability Statement:** All relevant data are within the paper.

**Funding:** This study was supported by the Ministry of Education, Science and Technological Development of the Republic of Serbia

## Abstract

### Background

The COVID-19 pandemic is currently one of the biggest public health threats for people's mental health. A particularly endangered group were students, who became highly affected by measures of social distance due to their active lifestyle. Therefore, the aim of this paper is to assess the level of self-reported stress, anxiety and depression of the student population in Serbia, in relation to demographic characteristics, living and studying conditions, students' activities during the epidemic, potential coronavirus infection and general, mental and physical health.

### Methods

We conducted a cross-sectional study of 580 undergraduate medical students from the University of Belgrade during the 2020/2021 school year. Mental health data were collected using the Depression Anxiety Stress Scales (DASS-21). Both bivariate and multivariate logistic regression analysis were used to examine the association between independent variables with the dependent variable mental health.

### Results

Women made up the majority of the sample with 80.3%. A total of 64.5%, 66.8% and 66.7% of students between the ages of 21 and 30 had severe depressive symptoms, severe degree of anxiety, and a severe degree of stress, respectively. Women almost twice as often (OR = 1.89) assessed their anxiety as severe and almost two and a half times more (OR = 2.39) perceived stress as severe compared to men. Students who lived with their families during studies two and a half times (OR = 2.57) more often assessed their stress as severe, compared to students who lived alone. Fifth- and sixth-year students were less likely to rate depression and anxiety as serious than the first-year students.

(Ministarstvo Prosvete, Nauke i Tehnološkog Razvoja RS, project No 200110). The funders had no role in study design, data collection and analysis, decision to publish, or preparation of the manuscript.

**Competing interests:** The authors have declared that no competing interests exist.

## Conclusions

Medical students reported their health as severely impaired in terms of depression, anxiety and stress reactions. The results indicate the need to launch a mental health program in the form of counseling and emotional support to students affected by the pandemic.

## Introduction

The world has confronted the COVID-19 outbreak for the first time in December 2019, when a new SARS-CoV-2 coronavirus was identified in the Chinese province of Hubei, in the city of Wuhan [1]. The pandemic scale of this novel coronavirus has led to the world's biggest public health crisis in over one hundred years and thus opened many unresolved issues. It has resulted in millions of dead people [2] and in the middle of 2020 there was an increase in COVID-19 cases among adults aged between 18 and 22 years [3].

As one of the main measures in the fight against this plague is physical distancing, along with the obligatory maintenance of hygienic measures and isolation from other people, the pandemic has caused a number of mental problems around the world. Italian researchers identified elevated levels of stress, anxiety and depression in the general population in response to the COVID-19 pandemic [4]. People with a history of psychiatric disorders were particularly affected during a pandemic [5], and their mental health is further impaired due to the long-term presence of the virus in the population. Also, single people showed severe symptoms of depression, while married people, especially females, showed high levels of anxiety [6].

The COVID-19 pandemic is currently one of the biggest public health threats and challenges for professionals' mental health [7]. It caused the closure or reorganization of most educational institutions around the world, including higher education institutions, and had a negative effect on students' mental health [8]. Among students in Italy, in addition to the concerns about the organization of studies and discomfort due to social distancing, results suggested that female students were more likely to complain of anxiety than male students [9]. A study from Serbia showed the negative impact of the COVID-19 disease pandemic on the psychological state of students [10]. Mental health problems and the occurrence of anxiety and affective emotional problems were also found in the student population of Slovakia [11]. A special problem was created among medical students who, due to the pandemic, could not have practical classes and contact with patients [12] and thus self-reported increased levels of anxiety and stress [6]. Their level of anxiety was even higher compared to the general population [13]. During the pandemic medical students from the Department of Public Health University in Northern New Jersey, US showed problems with mental health due to difficulty in coping with changed daily and student activities [14].

To our best knowledge, this is the first study in Serbia that examines the effects of the COVID-19 epidemic on students' mental health and the first study to observe its effects on the mental health of the medical student population.

Therefore, the aim of this paper is to assess the level of self-reported stress, anxiety, and depression of the student population in Serbia in relation to the demographic characteristics, living and studying conditions, students' activities during the epidemic, potential coronavirus infection, and general, mental and physical health.

## Method

### Study design and participants

This cross-sectional online survey included 580 medical students (who completely filled out the questionnaires) of integrated academic studies from the Medical Faculty University of

Belgrade during the 2020/2021 school year. The questionnaire was posted on the online teaching platform of the Faculty of Medicine called Reticulum, where students could fill it out in the period from January to March 2021.

The subjects were selected according to the principle of stratified cluster sample. The entire target population of full-time medical students on the integrated studies course was observed as one cluster, while students from each of the six years of study were observed as a separate group. Approximately, the same number of respondents was taken from each stratum. There were 105 first-year students, 85 second-year students, 93 third-year students, 84 fourth-year students, 79 fifth-year students, and 134 sixth-year students. The sample selected in this way is a representative sample, and the representativeness was determined on the basis of a sufficient number of observation units to assess the proportions, as well as a random selection of observation units, which enabled the generalization of the obtained results.

The research was conducted in accordance with the Declaration of Helsinki and approved by the Ethics Review Board of the Medical Faculty University of Belgrade by decision number 1322/III-22. All students gave written informed consent and after being informed about the purpose and procedure of the survey, they voluntarily started filling out anonymous questionnaires.

## Instruments and variables

In total, three questionnaires were created using the online survey tool entitled LimeSurvey. The general questionnaire and the health self-assessment questionnaire were constructed specifically for the purpose of this study, while the DASS-21 questionnaire was used to assess depression, anxiety, and stress.

The general questionnaire was designed to collect personal data related to the students' life before and during the COVID-19 epidemic. It collected data for eleven variables which, in addition to the demographic characteristics (gender–male/female; age; marital status–single/living with a partner; type of settlement–urban/rural; year of studies–I to VI and religious–yes/no) of the participants, also collected information about living and studying conditions before and after the outbreak of the COVID-19 epidemic (cohabitation during the studies—living alone/living with family or roommate; location during the state of emergency–in Belgrade/not in Belgrade or Serbia), information on the students' activities during the epidemic (online classes–few/moderate/plenty; problems with online learning platform–yes/no), and data on coronavirus infection (yes/no). The age variable was transformed into a categorical variable for research purposes ($< = 20$, 21–30, 31+).

Regarding the health self-assessment questionnaire the participants were asked to assess general, physical, and mental health on the Likert five-point scale (1-poor health, 5-good health). All three components of health were transformed into variables with three categories: poor, average, and good health.

The Depression Anxiety Stress Scales (DASS-21) is a questionnaire that was previously constructed and linguistically and culturally validated for the Serbian language, whose reliability measured by Cronbach's alpha was 0.92 [15]. It has 21 questions and three subscales that measure the level of depression, anxiety, and stress during the week before the survey [16]. Each subscale contains seven questions, and participants express their own degree of depression, anxiety and stress by selecting one of the four offered options (never, sometimes, often and always). Very severe depression was detected in subjects with a score of 28+, severe in subjects with a score of 21–27, moderate in subjects with a score of 14–20, mild in those with a score of 10–13, while a normal depressive reaction was in students with a score ≤9. Students with a score of 20–49 had very severe anxiety, those with a score of 15–19 had severe anxiety, those

with a score of 10–14 had moderate anxiety, and those with a score of 7–9 had mild anxiety, while normal anxiety was scored ≤ 6. Students with a score of 35–42 had a very severe stress reaction, 27–34 had severe stress, those with a score of 19–26 had moderate stress, those with a score of 11–18 had mild stress, while subjects with a score ≤10 had a normal stress reaction in the one-week period prior to testing. The score limits presented in this paper have been used in previous research [17]. For the purposes of this study, all three subscales were transformed to obtain variables with three categories: severe, moderate, and mild depression/anxiety/stress response.

## Statistical analysis

Data were described by descriptive statistics–categorical variables were reported as percentages (%) and frequencies (n), while means and standard deviation (SD) were calculated for numerical ones. The difference between the groups was analyzed using Chi-square test. Bivariate and multivariate logistic regressions were used to examine the correlation between independent and outcome variables and presented with odds ratio (OR) and confidence intervals (CIs). All variables were first included in the bivariate regression analyses, and then, regardless of whether they were statistically significant or not, included in the multivariate regression analyses (because of the possibility that might be a significant association between variables and because of their significance for the study topic according to the literature and authors' opinion). The level of statistical significance in all used analyses was set at p<0.05. Data were processed using the statistical package IBM SPSS V.24.0 (SPSS Inc. Chicago, Illinois, USA).

## Results

Table 1 shows the distribution of demographic variables, as well as variables related to both study and living conditions and the health status of the study participants.

They were predominantly female (80.3%), aged between 21 and 30 (68.4), religious (63.3%), and from urban areas (85.9%). More than one-third of participants (37.1%) had few organized online classes during the pandemic, 32.4% had problems with e-learning platforms, and 25.5% of students were infected with SARS-CoV-2 by the time of the survey. Out of all examined students, 71.9%, 66.7% and 50.5% self-perceived their general, physical and mental health as good, respectively. Severe symptoms of depression were recognized by 65.5% of students, 90.9% of students suffered from severe anxiety, and 85.6% were under severe stress.

A total of 64.5%, 66.8% and 66.7% of students between the ages of 21 and 30 reported severe depressive symptoms, severe degree of anxiety, and a severe stress, respectively (Table 2).

The percentage of women who perceived their anxiety as severe was 81.4%, while the percentage of women who perceived their stress as severe was 83.1%. The most severe symptoms of depression, anxiety, and stress were among students in the first (21.6%; 19.5% and 20.2%, respectively) and sixth (19.5%, 23.1%, and 21.8%, respectively) year of study, while the least severe symptoms occur in fifth-year students (10.8%, 12.5%, and 12.7%, respectively). Also, 60% of students with severe depressive symptoms were religious and 36.3% of them did not have adequate online learning platforms. Among students who self-reported severe depressive symptoms, 32.4% rated their mental health as good, and of those with severe anxiety, 65.1% considered their physical health to be good. Of all the participants who rated their general health as good, 69% of the participants self-reported severe stress.

The results of the bivariate logistic regressions are shown in Table 3.

Students between the ages of 21 and 30, as well as fifth and sixth-year students, were less likely to rate their depression as severe compared to students under the age of 20, and first-year students, respectively. The same pattern was observed for the self-assessment of anxiety.

**Table 1. Distribution of demographic variables, variables related to students medical studies and health status during the coronavirus epidemic, as well as, scales of depression, anxiety and stress.**

| Variables | n | % |
|---|---|---|
| Gender | | |
| Male | 114 | 19.7 |
| Female | 466 | 80.3 |
| Age | | |
| ≤20 | 174 | 30 |
| 21–30 | 397 | 68.4 |
| 31+ | 9 | 1.6 |
| Year of studies | | |
| First | 105 | 18.1 |
| Second | 85 | 14.7 |
| Third | 93 | 16 |
| Fourth | 84 | 14.5 |
| Fifth | 79 | 13.6 |
| Sixth | 134 | 23.1 |
| Marital status | | |
| Single | 456 | 78.6 |
| Married/Living with a partner | 124 | 21.4 |
| Religious | | |
| Yes | 367 | 63.3 |
| No | 213 | 36.7 |
| Settlement type | | |
| Urban | 498 | 85.9 |
| Rural | 82 | 14.1 |
| Cohabitation during the studies | | |
| Alone | 117 | 20.2 |
| With family/room mate | 463 | 79.8 |
| Location during the state of emergency | | |
| In Belgrade | 237 | 40.9 |
| Not in Belgrade/Serbia | 343 | 59.1 |
| Online classes | | |
| Few | 215 | 37.1 |
| Moderate | 235 | 40.5 |
| Plenty | 130 | 22.4 |
| Problems with online learning platform | | |
| Yes | 188 | 32.4 |
| No | 392 | 67.6 |
| Coronavirus infection | | |
| Yes | 148 | 25.5 |
| No/not sure | 432 | 74.5 |
| General health | | |
| Poor | 15 | 2.6 |
| Average | 148 | 25.5 |
| Good | 387 | 66.7 |
| Physical health | | |
| Poor | 34 | 5.9 |
| Average | 159 | 27.4 |

(*Continued*)

**Table 1.** (Continued)

| Variables | n | % |
|---|---|---|
| Good | 387 | 66.7 |
| Mental health | | |
| Poor | 94 | 16.2 |
| Average | 193 | 33.3 |
| Good | 293 | 50.5 |
| Depression | | |
| Mild | 0 | 0 |
| Moderate | 200 | 34.5 |
| Severe | 380 | 65.5 |
| Anxiety | | |
| Mild | 0 | 0 |
| Moderate | 53 | 9.1 |
| Severe | 527 | 90.9 |
| Stress | | |
| Mild | 20 | 3.4 |
| Moderate | 64 | 11 |
| Severe | 496 | |

On contrary, non-religious students were one and a half times more likely to experience a major depressive episode (OR = 1.52) compared to religious students. Students who did not have problems with online learning platforms were less likely to assess their depression as severe. Students who rated their mental health as average or good, as well as those who rated physical health as good, were less likely to experience a severe depressive reaction, compared to students who rated their physical and mental health as poor. Compared to men, women almost twice as often (OR = 1.89) and almost two and a half times more women (OR = 2.39) perceived their anxiety and stress as severe, respectively.

As indicated in Table 4, sixth-year (OR = 0.21) and fifth-year students (OR = 0.12) were less likely to perceive their depression as severe as first-year students. Students of the second (OR = 0.18), third (OR = 0.11), fifth, and sixth year (OR = 0.09) evaluated less frequently stressful reactions as severe, compared to students of the first year of medicine. Students who rated their mental health as good (OR = 0.01) or average (OR = 0.06) were also less likely to rate their depression as severe. Women more than twice (OR = 2.21) and more than three times (OR = 3.37) times more often assessed their anxiety or stress reactions as severe compared to men.

## Discussion

The current study indicates that 65.5%, 85.6%, and 90.9% of students self-reported severe depressive, stress, and anxiety reactions, respectively. Our results are in line with the study from Russia where students showed higher levels of depression and higher levels of stress during the pandemic, but the opposite compared to students from Belarus, where social distance measures were not effective in the first months of the pandemic [18]. A study conducted on students in Switzerland showed that depressive symptoms were present among more than a quarter of the student population [19]. Similar results, but with lower levels of depression, anxiety, and stress (42%, 44.5%, and 66.0%, respectively) were obtained in another study conducted among the general population in Serbia [10]. A possible explanation for higher levels in our study may be attributed to the population that participated. Namely, medical students and

**Table 2. Distribution of self-assessed depression, anxiety and stress, as well as their differences in relation to students demographic variables, their medical studies and health status during the coronavirus epidemic.**

| Variables | Depression Scale | | P-value* | Anxiety Scale | | P-value* | Stress Scale | | | P-value* |
|---|---|---|---|---|---|---|---|---|---|---|
| | Moderate (%) | Severe (%) | | Moderate (%) | Severe (%) | | Mild (%) | Moderate (%) | Severe (%) | |
| Gender | | | 0.417 | | | 0.043 | | | | <0.001 |
| Male | 21.5 | 18.7 | | 30.2 | 18.6 | | 45 | 32.8 | 16.9 | |
| Female | 78.5 | 81.3 | | 69.8 | 81.4 | | 55 | 67.2 | 83.1 | |
| Age | | | 0.003 | | | 0.020 | | | | 0.009 |
| ≤20 | 21.5 | 34.5 | | 13.2 | 31.7 | | 5 | 21.9 | 32.1 | |
| 21–30 | 76 | 64.5 | | 84.9 | 66.8 | | 95 | 73.4 | 66.7 | |
| 31+ | 2.5 | 1.1 | | 1.9 | 1.5 | | 0 | 4.7 | 1.2 | |
| Year of studies | | | <0.001 | | | 0.012 | | | | 0.016 |
| First | 11.5 | 21.6 | | 3.8 | 19.5 | | 0 | 7.8 | 20.2 | |
| Second | 13 | 15.5 | | 17 | 14.4 | | 5 | 17.2 | 14.7 | |
| Third | 14 | 17.1 | | 22.6 | 15.4 | | 30 | 17.2 | 15.3 | |
| Fourth | 12.5 | 15.5 | | 9.4 | 15 | | 10 | 9.4 | 15.3 | |
| Fifth | 19 | 10.8 | | 24.5 | 12.5 | | 30 | 15.6 | 12.7 | |
| Sixth | 30 | 19.5 | | 22.6 | 23.1 | | 25 | 32.8 | 21.8 | |
| Marital status | | | 0.490 | | | 0.640 | | | | 0.199 |
| Single | 77 | 79.5 | | 81.1 | 78.4 | | 90 | 84.4 | 77.4 | |
| Married/Living with a partner | 33 | 20.5 | | 18.9 | 21.6 | | 10 | 15.6 | 22.6 | |
| Religious | | | 0.024 | | | 0.890 | | | | 0.309 |
| Yes | 69.5 | 60 | | 64.2 | 63.2 | | 50 | 68.8 | 63.1 | |
| No | 30.5 | 40 | | 35.8 | 36.8 | | 50 | 31.2 | 36.9 | |
| Settlement type | | | 0.749 | | | 0.583 | | | | 0.156 |
| Urban | 86.5 | 85.5 | | 88.7 | 85.6 | | 90 | 78.1 | 86.7 | |
| Rural | 13.5 | 14.5 | | 11.3 | 14.1 | | 10 | 21.9 | 13.3 | |
| Cohabitation during the studies | | | 0.311 | | | 0.122 | | | | 0.002 |
| Alone | 22.5 | 18.9 | | 28.3 | 19.4 | | 35 | 34.4 | 17.7 | |
| With family/room mate | 77.5 | 81.1 | | 71.7 | 80.6 | | 65 | 65.6 | 82.3 | |
| Location during the state of emergency | | | 0.686 | | | 0.847 | | | | 0.713 |
| In Belgrade | 42 | 40.3 | | 39.6 | 41 | | 35 | 37.5 | 41.5 | |
| Not in Belgrade/Serbia | 58 | 59.7 | | 60.4 | 59 | | 65 | 62.5 | 58.5 | |
| Online classes | | | 0.429 | | | 0.275 | | | | 0.849 |
| Few | 38.5 | 36.3 | | 43.4 | 36.4 | | 30 | 39.1 | 37.1 | |
| Moderate | 37 | 42.4 | | 30.2 | 41.6 | | 50 | 35.9 | 40.7 | |
| Plenty | 24.5 | 21.3 | | 26.4 | 22 | | 20 | 25 | 22.2 | |
| Problems with online learning platform | | | 0.006 | | | 0.057 | | | | 0.251 |
| Yes | 25 | 36.3 | | 20.8 | 33.6 | | 30 | 23.4 | 33.7 | |
| No | 75 | 63.7 | | 79.2 | 66.4 | | 70 | 76.6 | 66.3 | |
| Coronavirus infection | | | 0.552 | | | 0.875 | | | | 0.539 |
| Yes | 27 | 24.7 | | 26.4 | 25.4 | | 15 | 25 | 26 | |
| No/not sure | 73 | 75.3 | | 73.6 | 74.5 | | 85 | 75 | 74 | |
| General health | | | <0.001 | | | 0.015 | | | | 0.002 |
| Poor | 0 | 3.9 | | 0 | 2.8 | | 0 | 0 | 3 | |
| Average | 10 | 33.7 | | 11.3 | 26.9 | | 0 | 14.1 | 28 | |
| Good | 90 | 62.4 | | 88.7 | 70.2 | | 100 | 85.9 | 69 | |
| Physical health | | | <0.001 | | | 0.029 | | | | 0.005 |

(*Continued*)

**Table 2.** (Continued)

| Variables | Depression Scale | | P-value* | Anxiety Scale | | P-value* | Stress Scale | | | P-value* |
|---|---|---|---|---|---|---|---|---|---|---|
| | Moderate (%) | Severe (%) | | Moderate (%) | Severe (%) | | Mild (%) | Moderate (%) | Severe (%) | |
| Poor | 1.5 | 8.2 | | 3.8 | 6.1 | | 0 | 0 | 6.9 | |
| Average | 14.5 | 34.2 | | 13.2 | 28.8 | | 5 | 21.9 | 29 | |
| Good | 84 | 57.6 | | 83 | 65.1 | | 95 | 78.1 | 64.1 | |
| Mental health | | | <0.001 | | | <0.001 | | | | <0.001 |
| Poor | 0.5 | 24.5 | | 0 | 17.8 | | 0 | 0 | 19 | |
| Average | 14.5 | 43.2 | | 13.2 | 35.3 | | 5 | 15.6 | 36.7 | |
| Good | 85 | 32.4 | | 86.8 | 46.9 | | 95 | 84.4 | 44.4 | |

*According to Chi-squre test

medical personnel were more than the general population in contact with patients, moved to places where the virus transmission was higher, and generally knew more about the virus than other people, which could have further compromised their mental health [20].

The results of our study showed that only 32.4% of students with self-reported severe depressive reactions had good mental health. In a study with Switzerland's students, depressive symptoms occurred more often in female students than in males, as well as in students who cared more about their health or already had some health problems [19]. In our study, no gender difference was found in terms of self-reported depressive reactions, although, a study conducted among Italian students [9] showed a significant difference between men and women, that is, women showed more depressive reactions.

Our study showed that first year students self-reported more severe depressive reactions than their older colleagues at the second, third, fifth, and sixth years of studies. Similar results were obtained in an extensive study conducted in nine countries by Ochnik et al. [21], which showed that students worldwide did not differ much in their reactions to the pandemic, but also that students in their younger years were more likely to self-report more depressive reactions. Students from our study who did not have problems with online learning platforms during the pandemic were 0.58 times less likely to self-report severe depressive reactions compared to students who had certain problems. For students who had problems with online learning platforms, professors from Belgrade Medical Faculty might organize additional consultations on weekly basis, so they can catch up on the material. Also, medical students who were non-religious self-reported severe depressive symptoms one and a half times more often (OR = 1.52) than religious students, which is in line with the findings of a study conducted with students in Malaysia [22] who generally turned to religion as a mechanism of coping stressful situations. Students from our study who aren't religious can be referred to attend courses at the Department of Humanities that include topics in religion (like Health Psychology) and the content like this may help them find some answers on how to cope with stress during the days of the pandemic.

The results of this study showed that anxiety in our students was present in 90.9%, which indicates the high vulnerability of Serbian students to pandemic conditions. A study conducted among Chinese medical students [12] showed that 25% of students had problems with anxiety due to the virus pandemic and its consequences on everyday life and education. Anxiety was also a reaction of US students to the pandemic and its uncertainties, such as financial problems, unemployment, social isolation, concerns related to study conditions, and the organization of distance learning [23]. In our study, female students were more likely to self-report anxiety, that is, 2.21 times more often than male students, which is in line with data from a

**Table 3. Associations of scales of depression, anxiety and stress with students' demographic variables, their medical studies and health status during the coronavirus pandemic–results of bivariate logistic regressions.**

| Variables | Bivariate logistic regression (Severe versus moderate) | | | | | |
|---|---|---|---|---|---|---|
| | Depression Scale | | Anxiety Scale | | Stress Scale | |
| | OR (95%CI) | *P*-value | OR (95%CI) | *P*-value | OR (95%CI) | *P*-value |
| Gender | | | | | | |
| Male | 1 | | 1 | | 1 | |
| Female | 1.19 (0.78–1.82) | 0.418 | 1.89 (1.01–3.54) | 0.046 | 2.39 (1.35–4.24) | 0.003 |
| Age | | | | | | |
| ≤20 | 1 | | 1 | | 1 | |
| 21–30 | 0.53 (0.35–0.79) | 0.002 | 0.33 (0.15–0.74) | 0.007 | 0.62 (0.33–1.16) | 0.135 |
| 31+ | 0.26 (0.07–1.02) | 0.054 | 0.34 (0.04–3.06) | 0.333 | 0.18 (0.04–0.78) | 0.022 |
| Year of studies | | | | | | |
| First | 1 | | 1 | | 1 | |
| Second | 0.64 (0.33–1.22) | 0.175 | 0.16 (0.03–0.78) | 0.023 | 0.33 (0.11–0.99) | 0.049 |
| Third | 0.65 (0.34–1.23) | 0.189 | 0.13 (0.03–0.60) | 0.009 | 0.34 (0.12–1.04) | 0.058 |
| Fourth | 0.66 (0.34–1.28) | 0.219 | 0.30 (0.06–1.62) | 0.164 | 0.63 (0.19–2.15) | 0.464 |
| Fifth | 0.30 (0.16–0.57) | <0.001 | 0.10 (0.02–0.45) | 0.003 | 0.04 (0.10–0.96) | 0.043 |
| Sixth | 0.35 (0.19–0.61) | <0.001 | 0.20 (0.04–0.90) | 0.036 | 0.01 (0.09–0.71) | 0.009 |
| Marital status | | | | | | |
| Single | 1 | | 1 | | 1 | |
| Married/Living with a partner | 0.86 (0.57–1.30) | 0.490 | 1.19 (0.58–2.44) | 0.640 | 1.58 (0.78–3.19) | 0.208 |
| Religious | | | | | | |
| Yes | 1 | | 1 | | 1 | |
| No | 1.52 (1.05–2.19) | 0.024 | 1.04 (0.58–1.88) | 0.890 | 1.29 (0.74–2.25) | 0.378 |
| Settlement type | | | | | | |
| Urban | 1 | | 1 | | 1 | |
| Rural | 1.08 (0.66–1.78) | 0.749 | 1.32 (0.55–3.19) | 0.538 | 0.55 (0.29–1.05) | 0.069 |
| Cohabitation during the studies | | | | | | |
| Alone | 1 | | 1 | | 1 | |
| With family/room mate | 1.24 (0.82–1.89) | 0.311 | 1.64 (0.87–3.10) | 0.125 | 2.43 (1.38–4.27) | 0.002 |
| Location during the state of emergency | | | | | | |
| In Belgrade | 1 | | 1 | | 1 | |
| Not in Belgrade/Serbia | 1.07 (0.75–1.52) | 0.686 | 0.95 (0.53–1.68) | 0.847 | 0.85 (0.49–1.45) | 0.538 |
| Online classes | | | | | | |
| Few | 1 | | 1 | | 1 | |
| Moderate | 1.24 (0.82–1.80) | 0.332 | 1.64 (0.84–3.19) | 0.146 | 1.19 (0.66–2.18) | 0.564 |
| Plenty | 0.92 (0.59–1.45) | 0.726 | 0.99 (0.49.2.00) | 0.983 | 0.93 (0.48–1.83) | 0.842 |
| Problems with online learning platform | | | | | | |
| Yes | 1 | | 1 | | 1 | |
| No | 0.58 (0.40–0.86) | 0.006 | 0.52 (0.26–1.03) | 0.061 | 0.60 (0.33–1.11) | 0.103 |
| Coronavirus infection | | | | | | |
| Yes | 1 | | 1 | | 1 | |
| No/not sure | 1.12 (0.76–1.66) | 0.552 | 1.05 (0.55–2.00) | 0.875 | 0.95 (0.52–1.73) | 0.862 |
| General health | | | | | | |
| Poor | 1 | | 1 | | 1 | |
| Average | / | 0.999 | / | 0.999 | / | 0.999 |
| Good | / | 0.998 | / | 0.999 | / | 0.999 |
| Physical health | | | | | | |

(*Continued*)

**Table 3.**  (Continued)

| Variables | Bivariate logistic regression (Severe versus moderate) | | | | | |
|---|---|---|---|---|---|---|
| | Depression Scale | | Anxiety Scale | | Stress Scale | |
| | OR (95%CI) | *P*-value | OR (95%CI) | *P*-value | OR (95%CI) | *P*-value |
| Poor | 1 | | 1 | | 1 | |
| Average | 0.43 (0.12–1.52) | 0.191 | 1.36 (0.27–6.84) | 0.711 | / | 0.998 |
| Good | 0.13 (0.04–0.42) | 0.001 | 0.49 (0.11–2.20) | 0.335 | / | 0.998 |
| Mental health | | | | | | |
| Poor | 1 | | 1 | | 1 | |
| Average | 0.06 (0.01–0.45) | 0.006 | / | 0.997 | / | 0.996 |
| Good | 0.01 (0.01–0.06) | <0.001 | / | 0.996 | / | 0.996 |

study conducted in New York [24]. Anxiety also occurred among adolescents and students during the pandemic in Pakistan [25], and the possible cause may be complete isolation, which was especially difficult for students who are by their nature very active, curious, and friendly. Medical students from Jordan [8], showed a higher degree of anxiety during the pandemic due to the way the studies were organized, that is, the difficulty of organizing primarily practical classes.

In our study, 17.8% of students stated that they have poor mental health, and self-perceived poor mental health was a predictor of higher levels on the anxiety scale. Research with students from Bangladesh showed that mental health problems can predict some psychopathological symptoms [26] and that there is a link between fear and anxiety about the virus and anxiety reactions. High anxiety levels, were also found in medical students from Mexico [27], and consequently, the students stated that anxiety and depressive reactions in that period were the main problems in normal everyday life. Health-related anxiety also occurred in students in Germany [28], where half of them reported increased concerns during the COVID-19 pandemic. Students from Serbia self-reported more intense and more often anxiety reactions at the beginning of their studies compared to those of older students and such findings are consistent with results in some other countries such as Slovenia, Germany, Turkey or Israel [21].

Our findings showed that 85.6% of medical students from Serbia self-reported severe stress due to the SARS-CoV-2 virus pandemic and 3.37 times more often female medical students compared to their male counterparts, which is in line with the student population in China [29]. At the beginning of their studies, students more often self-reported severe stress reactions compared to their older colleagues, which was also confirmed in a study by Ochnik et al. [21]. Our study did not confirm that social support in family life has a protective effect on mental health. Students who lived with their family were two and a half times (OR = 2.57) more likely to self-report severe stress in relation to their colleagues who lived alone and perceived their stress as moderate or mild. On the contrary, research conducted in Thailand [30] showed that stress reduction is more successful if students are provided with some form of social support and assistance. The transition of society, the lack of social support within some families, and the value system that has reigned in Serbia recently might be partly responsible for the ongoing situation, while in countries with a very expressed collectivist culture, such as Indochina, the value of social support has not declined.

During the pandemic, the University did not provide any organized activities of assistance to students. The students themselves organized support groups through social networks. The primary focus of the University was the organization of online classes while taking care of students' mental health was of secondary importance.

**Table 4. Associations of scales of depression, anxiety and stress with students' demographic variables, their medical studies and health status during the coronavirus pandemic–results of multivariate logistic regressions.**

| Variables | Multivariate logistic regression (Severe versus moderate) | | | | | |
| --- | --- | --- | --- | --- | --- | --- |
| | Depression Scale | | Anxiety Scale | | Stress Scale | |
| | OR (95%CI) | P-value | OR (95%CI) | P-value | OR (95%CI) | P-value |
| Gender | | | | | | |
| Male | 1 | | 1 | | 1 | |
| Female | 1.56 (0.91–2.66) | 0.107 | 2.21 (1.09–4.49) | 0.027 | 3.37 (1.70–6.68) | <0.001 |
| Age | | | | | | |
| ≤20 | 1 | | 1 | | 1 | |
| 21–30 | 1.81 (0.61–5.44) | 0.288 | 0.59 (0.14–2.47) | 0.467 | 1.82 (0.42–7.90) | 0.424 |
| 31+ | 0.37 (0.04–3.20) | 0.370 | 0.31 (0.02–5.10) | 0.416 | 0.21 (0.02–2.20) | 0.194 |
| Year of studies | | | | | | |
| First | 1 | | 1 | | 1 | |
| Second | 0.53 (0.23–1.25) | 0.150 | 0.20 (0.03–1.14) | 0.069 | 0.18 (0.05–0.69) | 0.012 |
| Third | 0.32 (0.09–1.19) | 0.090 | 0.16 (0.02–1.40) | 0.099 | 0.11 (0.02–0.71) | 0.021 |
| Fourth | 0.26 (0.06–1.07) | 0.061 | 0.42 (0.04–4.25) | 0.461 | 0.18 (0.02–1.46) | 0.109 |
| Fifth | 0.12 (0.03–0.49) | 0.003 | 0.16 (0.02–1.49) | 0.107 | 0.09 (0.01–0.71) | 0.023 |
| Sixth | 0.21 (0.05–0.81) | 0.023 | 0.36 (0.04–3.32) | 0.366 | 0.09 (0.01–0.71) | 0.022 |
| Marital status | | | | | | |
| Single | 1 | | 1 | | 1 | |
| Married/Living with a partner | 1.05 (0.62–1.77) | 0.868 | 1.34 (0.61–2.98) | 0.466 | 2.01 (0.90–4.48) | 0.088 |
| Religious | | | | | | |
| Yes | 1 | | 1 | | 1 | |
| No | 1.50 (0.95–2.36) | 0.083 | 1.10 (0.56–2.14) | 0.786 | 1.36 (0.71–2.62) | 0.359 |
| Settlement type | | | | | | |
| Urban | 1 | | 1 | | 1 | |
| Rural | 1.33 (0.73–2.43) | 0.358 | 1.40 (0.54–3.62) | 0.494 | 0.57 (0.27–1.20) | 0.140 |
| Cohabitation during the studies | | | | | | |
| Alone | 1 | | 1 | | 1 | |
| With family/room mate | 1.18 (0.70–2.01) | 0.531 | 1.53 (0.74–3.16) | 0.253 | 2.57 (1.31–5.03) | 0.006 |
| Location during the state of emergency | | | | | | |
| In Belgrade | 1 | | 1 | | 1 | |
| Not in Belgrade/Serbia | 1.06 (0.68–1.66) | 0.795 | 0.90 (0.46–1.75) | 0.748 | 0.99 (0.52–1.89) | 0.938 |
| Online classes | | | | | | |
| Few | 1 | | 1 | | 1 | |
| Moderate | 0.96 (0.58–1.61) | 0.892 | 1.64 (0.76–3.56) | 0.206 | 1.03 (0.50–2.13) | 0.938 |
| Plenty | 0.55 (0.28–1.07) | 0.077 | 0.83 (0.33–2.06) | 0.687 | 0.52 (0.21–1.29) | 0.157 |
| Problems with online learning platform | | | | | | |
| Yes | 1 | | 1 | | 1 | |
| No | 0.75 (0.47–1.22) | 0.247 | 0.68 (0.32–1.48) | 0.335 | 0.94 (0.46–1.90) | 0.857 |
| Coronavirus infection | | | | | | |
| Yes | 1 | | 1 | | 1 | |
| No/not sure | 0.88 (0.54–1.44) | 0.617 | 0.99 (0.48–2.02) | 0.974 | 0.80 (0.40–1.60) | 0.530 |
| General health | | | | | | |
| Poor | 1 | | 1 | | 1 | |
| Average | / | 0.998 | / | 0.998 | / | 0.999 |
| Good | / | 0.998 | / | 0.998 | / | 0.999 |
| Physical health | | | | | | |

(*Continued*)

**Table 4.** (Continued)

| Variables | Multivariate logistic regression (Severe versus moderate) | | | | | |
| --- | --- | --- | --- | --- | --- | --- |
| | Depression Scale | | Anxiety Scale | | Stress Scale | |
| | OR (95%CI) | *P*-value | OR (95%CI) | *P*-value | OR (95%CI) | *P*-value |
| Poor | 1 | | 1 | | 1 | |
| Average | 1.23 (0.26–5.92) | 0.795 | 3.79 (0.52–27.49) | 0.187 | / | 0.998 |
| Good | 0.71 (0.14–3.57) | 0.683 | 2.80 (0.36–21.68) | 0.324 | / | 0.998 |
| Mental health | | | | | | |
| Poor | 1 | | 1 | | 1 | |
| Average | 0.06 (0.01–0.50) | 0.009 | / | 0.996 | / | 0.996 |
| Good | 0.01 (0.01–0.07) | <0.001 | / | 0.996 | / | 0.996 |

In Belgrade, the students can get specialist consultative services (covered by health insurance) for the protection of their mental health at the Institute for students' healthcare as primary health care institution. There is no program or center for psychological counseling at Belgrade University so the creation of a network of centers for counseling and psychological help for students as an integrated part of the health system is of crucial importance for the prevention and protection of mental health, but also for young people in general. The individual, group, and workshop activities for students could be provided in those centers by teachers, assistants, final year students, psychotherapists, and other psychologists, but also experts and researchers from numerous other institutions. Basic activities would be direct counseling, peer counseling, SOS telephone service, internet counseling (through an online Faculty platform), organization of forums, seminars, round tables, the realization of empirical research, training and education, etc. Also, if one student needs hospital treatment, professional help would be provided. The support program should be available to students throughout the year and could start at the beginning of the academic year. For those students with poor mental health it must take into consideration the specificities of the population to which it refers and all other contextual specificities [31]. During the orientation week or mental health gatekeeper training, younger students can learn from professors or senior colleagues how to recognize the first signs of mental problems, how to successfully deal with stress and use coping mechanisms, and where to find professional help [32, 33].

According to our best knowledge, this is the first study in Serbia that examines the effects of the COVID-19 pandemic on students' mental health and the first study to observe the mental health of the medical student population. Also, the advantage of this study was the random sampling process, and consequently representative sample. However, there are several limitations that need to be stressed. The generalization of results cannot be applied to students of other faculties of the University of Belgrade or to students in the rest of Serbia. Due to the epidemiological situation, the study was conducted online, which could have led to uneven conditions for completing the questionnaire. This was partially avoided by detailed written instructions and those related to the conditions in which the completion of the questionnaire should be organized, but still, we assume that some of the participants may have ignored these instructions. Students could also be dishonest in their responses.

## Conclusion

As a consequence of the SARS-CoV-2 virus pandemic, students attending undergraduate medical studies in Belgrade developed mental health problems. The majority of students assessed their anxiety and depressive reactions as moderate or severe, while stress reactions were

sometimes assessed as mild, but more often as moderate or severe. The COVID-19 pandemic had a great impact on the overall mental health of medical students and therefore it is important to pay attention to preserving their mental health and to launch programs that would provide, primarily social support and assistance in the form of consultative work.

## Supporting information

**S1 Table.**
(PDF)

## Author Contributions

**Conceptualization:** Nikola Mirilović, Janko Janković.

**Data curation:** Nikola Mirilović.

**Formal analysis:** Nikola Mirilović, Janko Janković.

**Methodology:** Nikola Mirilović, Janko Janković, Milan Latas.

**Supervision:** Janko Janković, Milan Latas.

**Writing – original draft:** Nikola Mirilović, Janko Janković.

**Writing – review & editing:** Nikola Mirilović, Janko Janković, Milan Latas.

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
