## [Decision Letter · Decision Letter 0]

14 Jul 2022

PONE-D-22-11573The impact of the COVID-19 epidemic on students’ mental health: a cross-sectional studyPLOS ONE

Dear Dr. Janković,

Thank you for submitting your manuscript to PLOS ONE. After careful consideration, we feel that it has merit but does not fully meet PLOS ONE’s publication criteria as it currently stands. Therefore, we invite you to submit a revised version of the manuscript that addresses the points raised during the review process.

We look forward to receiving your revised manuscript.

Kind regards,

Dr Aleksandra Barac

Academic Editor

PLOS ONE

Journal Requirements:

2. In the ethics statement in the Methods and online submission information, please ensure that you have specified (1) whether consent was informed and (2) what type you obtained (for instance, written or verbal, and if verbal, how it was documented and witnessed). If your study included minors, state whether you obtained consent from parents or guardians. If the need for consent was waived by the ethics committee, please include this information.

Reviewers' comments:

Reviewer's Responses to Questions

**Comments to the Author**

1. Is the manuscript technically sound, and do the data support the conclusions?

Reviewer #1: Partly

Reviewer #2: Yes

Reviewer #3: Yes

2. Has the statistical analysis been performed appropriately and rigorously? 

Reviewer #1: No

Reviewer #2: Yes

Reviewer #3: Yes

3. Have the authors made all data underlying the findings in their manuscript fully available?

Reviewer #1: Yes

Reviewer #2: Yes

Reviewer #3: Yes

4. Is the manuscript presented in an intelligible fashion and written in standard English?

Reviewer #1: No

Reviewer #2: Yes

Reviewer #3: Yes

5. Review Comments to the Author

Reviewer #1: Reviewer one

The article of self-reported experiences of depression, stress and anxiety among medical students in the University of Belgrade, Serbia is well-written, but it is generally confusing and needs refining, if it is to be published in this journal. The issues I will raise are conceptual, methodological and analytical.

CONCEPTUAL AND METHODOLOGICAL

The authors say they applied stratified random sampling. It is rare indeed to have medical schools with more Female students. But if University of Belgrade, Serbia has more female students, one of the purposes of stratified random sampling would be to ‘balance-out’ this gender anomaly. Therefore, to have a sample – with approximately four times female students, makes it superfluous to consider gender as an analytical tool . To put it in another way, given this huge gender imbalance, authors could have just retained female students’ self-reported experiences through out the article.

ANALYTICAL

Only students in clinical years ( 4 , 5, 6 years) come in contact with patients. Therefore attributing the high figures of self-reported anxiety, stress and depression in medical students during covid-19 to interacting with patients is confusing. It is even more confusing that students in senior years (4, 5, 6 years), self-reported being in good mental health.

There is always a big difference between self-report and exact experience in clinical categories in mental health. The authors instead use the terms self-report and ‘experience’ inter-changeably. I think what was assessed is self-report of stress, anxiety and depression.

If indeed up to 32.4% experienced, severe depression reactions, authors need to state clearly how they collected data with students who had symptoms including suicidal thoughts, lack of focus, and losing interest in pleasurable activities such as responding to lengthy online questionnaire.

Were these students who ‘experienced’ severe depression, treat by technical people. Severe depression -as-a-clinical category may require medication and Not Counselling. If the client had a symptom like suicidal thoughts, that is an emergency, requiring admission and close monitoring – not counselling and guidance.

Some sentences are poorly written. For example, a sentence like this lacks clarity. 71.9%, 66.7%. and 50.5% of students perceived their general, physical and mental health are good respectively. And yet, also 64.%%, 60.7% and 66.7% of students of ages 20 to 30 experienced severe depression symptoms, severe anxiety and severe stress. Please clarify how this contradiction can exists.

And if there were co-morbidity of severe depression and severe anxiety; this amounts to an emergency in mental health. Did you attend to these students, by referring them to a clinician?

In mental health, one needs to distinguish between commonly used words like stress, anxiety and depression – and sometimes schizophrenia, yet they are at the same time clinical categories with specific symptoms- over-a-six months period . In our experience, students in pre-clinical years, (first through to third year students) – use these clinical terms casually. I say casually because they do not exactly understand what they mean,

It is not surprising that on page 32, for instance, first year students, or generally students in junior year of study report moderate to severe depression than older colleagues .

Reviewer #2: In the statistical analysis section the sentence , "All predictors were first included in the bi-variate regression analyses, and then, regardless of whether they were statistically significant or not, included in the multivariate regression analyses". why you entered all variables into the multi-variable regression analysis without filtering out from the bi-variate analysis?

Reviewer #3: The impact of the COVID-19 epidemic on students' mental health: a cross-sectional study

Thank you for allowing me to review this study. This interesting study targets the most important concern among students, particularly medical students: mental health status and the emotional reaction to the pandemic.

- The significance of the study was clearly described in general and within the context of medical students and the country.

- The introduction was informative regarding all supportive global, international, national, and local literature.

- The study's purpose reflected the significance and context of the study.

- Methods were explained with some details (sampling methods, instruments, ethical concerns, data collection procedure, and the types of analysis).

- The results are presented clearly in tables and text.

- The discussion discussed the results within the context of the literature, the current situation, and the context of education and medical students.

- The authors highlighted the possible limitations and proposed a solution.

Minor changes:

- Typo error line 206 (form should be from).

- In the discussion part, you need to mention if there are any programs at all levels already run in the universities in Belgrade and if there were any activities during the pandemic for students.

- The conclusion must mention the differences in mental health between students to be considered in any designed program.

- Study implications in clinical practice, policy, and research should be considered and added. What type of programs should be launched, when, and how.

- Authors can suggest some recommendations specifically for the current sample, taking the factors associated with their mental health such as religion, study level, gender, access to online platforms..etc). for example, you may have specific actions for younger students during orientation week, such so.

6. PLOS authors have the option to publish the peer review history of their article (what does this mean?). If published, this will include your full peer review and any attached files.

Reviewer #1: **Yes: **Grace Akello, PhD

Reviewer #2: **Yes: **Ayechew Ademas

Reviewer #3: No

---

## [Author Response · Author response to Decision Letter 0]

7 Sep 2022

PONE-D-22-11573

The impact of the COVID-19 epidemic on students’ mental health: a cross-sectional study

PLOS ONE

Response to the Reviewer’s Comments

Reviewer #1: The article of self-reported experiences of depression, stress and anxiety among medical students in the University of Belgrade, Serbia is well-written, but it is generally confusing and needs refining, if it is to be published in this journal. The issues I will raise are conceptual, methodological and analytical.

COMMENTS

CONCEPTUAL AND METHODOLOGICAL

1.- The authors say they applied stratified random sampling. It is rare indeed to have medical schools with more Female students. But if University of Belgrade, Serbia has more female students, one of the purposes of stratified random sampling would be to ‘balance-out’ this gender anomaly. Therefore, to have a sample – with approximately four times female students, makes it superfluous to consider gender as an analytical tool . To put it in another way, given this huge gender imbalance, authors could have just retained female students’ self-reported experiences through out the article.

RESPONSE:

We appreciate your comment. It is not rare in this case, that is, at Medical Faculty University of Belgrade to have more female students than male students. It is a common gender proportion for Belgrade Medical Faculty on an annual basis, for each enrolled generation of students. So, we don't think in our case that this is a gender anomaly and our sample is a representative sample of medical students in Belgrade. We were interested in mental health of the entire student population, not in a stratified sample by gender. The subjects were selected according to the principle of stratified cluster sample. The entire population of medical students was observed as one cluster, while students from each of the six years of study were observed as a separate group. The representativeness was determined on the basis of a sufficient number of observation units to assess the proportions, as well as random selection of observation units, which enabled the generalization of the obtained results.

ANALYTICAL

2.- Only students in clinical years (4, 5, 6 years) come in contact with patients. Therefore attributing the high figures of self-reported anxiety, stress and depression in medical students during covid-19 to interacting with patients is confusing. It is even more confusing that students in senior years (4, 5, 6 years), self-reported being in good mental health.

RESPONSE:

Our Medical Faculty University of Belgrade is obviously a different case and students from all study years come in contact with patients. According to the curriculum first year students have obligatory module: ”The basics of clinical practice I” and second year students have obligatory module: “The basics of clinical practice II” with Physician in the community part when they go to Hospitals and Primary Health Care Centers and talk with the patients. Third year students have obligatory module: ”Clinical Propedeutics”. 

As you said in one of your comments (number 8) younger students use clinical terms like stress, anxiety and depression casually so based on our experience the finding that first year students had more severe depressive reactions than their older colleagues is not surprising or confusing at all.

Our findings of good mental health among older students are in line with data from other studies. For example, study conducted in nine countries showed that students in their younger years were more likely to experience more depressive reactions (Ochnik, D., Rogowska, A. M., Kuśnierz, C., et al. 2021. Mental health prevalence and predictors among university students in nine countries during the COVID-19 pandemic: A cross-national study. Scientific reports, 11 (1), pp. 1-13). Other study showed that students in their younger years were more likely to experience more anxiety reactions (Saeed, N. and Javed, N. 2021. Lessons from the COVID-19 pandemic: Perspectives of medical students. Pakistan Journal of Medical Sciences, 37 (5), pp. 1402-1407). 

3.- There is always a big difference between self-report and exact experience in clinical categories in mental health. The authors instead use the terms self-report and ‘experience’ inter-changeably. I think what was assessed is self-report of stress, anxiety and depression.

RESPONSE:

We totally agree with your comment. In our study we used The Depression Anxiety Stress Scales (DASS-21) questionnaire to assess depression, anxiety and stress. We used only self-reported measures and accordingly word ‘experience’ is changed with the words ‘self-report’ or ‘self-perceive’ throughout the whole text.

4.- If indeed up to 32.4% experienced, severe depression reactions, authors need to state clearly how they collected data with students who had symptoms including suicidal thoughts, lack of focus, and losing interest in pleasurable activities such as responding to lengthy online questionnaire.

RESPONSE:

It is clearly described under the methodological section how was the level of depression, anxiety and stress measured. Also, students reported on their self-assessment of mental health. There is nowhere in the text that up to 32.4% experienced, severe depression reactions. The truth is that among students who self-reported severe depressive symptoms, 32.4% rated their mental health as good. Students didn’t report on their diagnoses or clinical conditions. They reported exclusively on their own reactions to the pandemic and on their subjective, intrapersonal perceptions of their own condition. We did not use questions regarding suicidal thoughts, lack of focus, and losing interest in pleasurable activities.

5.- Were these students who ‘experienced’ severe depression, treat by technical people. Severe depression -as-a-clinical category may require medication and Not Counseling. If the client had a symptom like suicidal thoughts, that is an emergency, requiring admission and close monitoring – not counselling and guidance.

RESPONSE:

Students who reported severe depression were not treated by technical people. The main reason is that students filled out questionnaires anonymously so we don't know who are these students. As we have already said in our research we did not use questions regarding suicidal thoughts.

We agree with you that severe depression as a clinical category may require medication and treatment by technical people, but in this research, the DASS-21 self-reported questionnaire was used as a quantitative measure of distress along the 3 axes of depression, anxiety and stress. It is not a categorical measure of clinical diagnoses and the term severe depression doesn’t refer to the clinical manifestation of depression as a disorder. It is clearly explained under the methodological section how we obtained variables with three categories: severe, moderate, and mild depression/anxiety/stress. 

Counseling is the first step in the process of helping students who report mental health problems. If a student needs hospitalization and close monitoring he must be referred for medical treatment. Under the discussion section we proposed certain actions and according to reviewer#3 comments study implications in clinical practice, policy, and research were added. 

6.- Some sentences are poorly written. For example, a sentence like this lacks clarity. 71.9%, 66.7%. and 50.5% of students perceived their general, physical and mental health are good respectively. And yet, also 64.%%, 60.7% and 66.7% of students of ages 20 to 30 experienced severe depression symptoms, severe anxiety and severe stress. Please clarify how this contradiction can exists.

RESPONSE:

The sentences were reworded in the text. In the first sentence, we presented the answers of all students who self-assessed their general, physical and mental health. This self-assessment questionnaire collected data on how respondents generally assess three mentioned components of their health. In the second sentence, we presented the answers of students from the only one age category (20-30 years). These data were collected using the DASS-21 questionnaire and show self-report of depressive, anxiety and stress reactions. Therefore, these two sentences show different aspects of students' self-report of health. 

7.- And if there were co-morbidity of severe depression and severe anxiety; this amounts to an emergency in mental health. Did you attend to these students, by referring them to a clinician?

RESPONSE:

The answer on this question is similar as the answer on question number 5. In this research, the DASS-21 self-reported questionnaire was used and it is clearly explained under the methods section how three subscales were transformed to obtain variables with three categories: severe, moderate, and mild depression/anxiety/stress. The DASS-21 is a quantitative measure of distress along the 3 axes of depression, anxiety and stress. It is not a categorical measure of clinical diagnoses, so the term severe depression doesn’t refer to the clinical manifestation of depression as a disorder. We did not examine co-morbidity and students filled out questionnaires anonymously so we don't know specifically who reported severe depression or severe anxiety and therefore it was not possible to approach them.

8.- In mental health, one needs to distinguish between commonly used words like stress, anxiety and depression – and sometimes schizophrenia, yet they are at the same time clinical categories with specific symptoms- over-a-six months period. In our experience, students in pre-clinical years, (first through to third year students) – use these clinical terms casually. I say casually because they do not exactly understand what they mean. It is not surprising that on page 32, for instance, first year students, or generally students in junior year of study report moderate to severe depression than older colleagues .

RESPONSE:

That's right. We agree with you. Many terms are used incorrectly and to describe conditions that doesn’t indicate a diagnosis in everyday speech. For us it is also not surprising the finding that first year students had more severe depressive reactions than their older colleagues.

Reviewer #2:

1.- In the statistical analysis section the sentence, "All predictors were first included in the bi-variate regression analyses, and then, regardless of whether they were statistically significant or not, included in the multivariate regression analyses". why you entered all variables into the multi-variable regression analysis without filtering out from the bi-variate analysis?

RESPONSE:

Thank you for your comment. All variables were included in the multivariate regression analyses because of the possibility that might be a significant association between certain variables and because of their significance for the study topic according to the literature and authors' opinion. A part of the sentence was added in the method section of the manuscript, under statistical analysis subheading. Nevertheless, multivariate logistic regression, which was performed only with variables that were significant predictors in bivariate logistic regression, showed exactly the same results (significant predictors are the same variables) with our original analysis, that is, when all variables were included in the multivariate regression analyses despite their significance in the bivariate regression analyses. Please, see attached tables (we performed additional statistical analysis). When all variables were included in the analysis, the only difference is in more significant categories of the "years of study" variable. 

Reviewer #3: Thank you for allowing me to review this study. This interesting study targets the most important concern among students, particularly medical students: mental health status and the emotional reaction to the pandemic.

- The significance of the study was clearly described in general and within the context of medical students and the country.

- The introduction was informative regarding all supportive global, international, national, and local literature.

- The study's purpose reflected the significance and context of the study.

- Methods were explained with some details (sampling methods, instruments, ethical concerns, data collection procedure, and the types of analysis).

- The results are presented clearly in tables and text.

- The discussion discussed the results within the context of the literature, the current situation, and the context of education and medical students.

- The authors highlighted the possible limitations and proposed a solution.

Minor changes:

1.- Typo error line 206 (form should be from).

RESPONSE:

The typo error is corrected.

2.- In the discussion part, you need to mention if there are any programs at all levels already run in the universities in Belgrade and if there were any activities during the pandemic for students.

RESPONSE:

Thank you for your comment. One paragraph was added in the discussion part about the existence of programs and activities for students.

During the pandemic, the University did not provide any organized activities of assistance to students. The students themselves organized support groups through social networks. The primary focus of the University was the organization of online classes while taking care of students' mental health was of secondary importance.

The students can get specialist consultative services (covered by health insurance) for the protection of their mental health at the Institute for students’ healthcare as primary health care institution in Belgrade. But, there is no program or center for psychological counseling at Belgrade University. 

3.- The conclusion must mention the differences in mental health between students to be considered in any designed program.

RESPONSE:

Thank you for your comment. One sentence was added in the discussion part about the differences in mental health between students to be considered in any designed program.

Designed programs for providing support to students will take into account all the differences and specificities between students, primarily the level of depressive, anxiety and stress reactions.

4.- Study implications in clinical practice, policy, and research should be considered and added. What type of programs should be launched, when, and how.

RESPONSE:

Study implications were added in one paragraph under the discussion section. 

Regarding the fact that there is no program or center for psychological counseling at Belgrade University, we think that the creation of a network of centers for counseling and psychological help for students as an integrated part of the health system is of crucial importance for the prevention and protection of mental health, but also for young people in general. The individual, group and workshop activities for students could be provided in those centers by teachers, assistants, final year students, psychotherapists and other psychologists, but also experts and researchers from numerous other institutions. Basic activities would be direct counseling, peer counseling, SOS telephone service, internet counseling (through an online Faculty platform), organization of forums, seminars, round tables, the realization of empirical research, training and education, etc. Also, if one student needs hospital treatment, professional help would be provided. The support program should be available to students throughout the year and could start at the beginning of the academic year. 

5.- Authors can suggest some recommendations specifically for the current sample, taking the factors associated with their mental health such as religion, study level, gender, access to online platforms..etc). for example, you may have specific actions for younger students during orientation week, such so.

RESPONSE: 

We agree with your comment and suggestion. We added specific recommendations in a few sentences under the discussion section.

Programs for providing assistance to students with poor mental health must really take into consideration the specificities of the population to which it refers and all other contextual specificities. (Singh, S., Roy, D., Sinha, K., Parveen, S., Sharma, G., et al. Joshi, G. 2020. Impact of COVID-19 and lockdown on mental health of children and adolescents: A narrative review with recommendations. Psychiatry research, 293, pp. 113429). During the orientation week or mental health gatekeeper training, younger students can learn from professors or older colleagues how to successfully deal with stress and use coping mechanisms. Also, senior colleagues can show them how to recognize the first signs of mental problems and where they can get professional help. (Cvjetković, SJ. 2020. Research on predictors of psychological distress and willingness to seek professional psychological help among Belgrade University students. Doctoral Dissertation (in Serbian). Faculty of Medicine University of Belgrade). For students who had problems with online learning platforms, professors might organize additional consultations on weekly basis, so they can catch up on the material. Students who aren’t religious can be referred to attend courses at the Department of Humanities that include topics in religion (like Health Psychology) and the content like this may help them find some answers on how to cope with stress during the days of the pandemic.

---

## [Editor Report · Decision Letter 1]

12 Sep 2022

The impact of the COVID-19 epidemic on students’ mental health: a cross-sectional study

PONE-D-22-11573R1

Dear Dr. Janković,

We’re pleased to inform you that your manuscript has been judged scientifically suitable for publication and will be formally accepted for publication once it meets all outstanding technical requirements.

Kind regards,

Aleksandra Barac

Academic Editor

PLOS ONE

---

## [Editor Report · Acceptance letter]

14 Sep 2022

PONE-D-22-11573R1 

The impact of the COVID-19 epidemic on students’ mental health: a cross-sectional study 

Dear Dr. Janković:

I'm pleased to inform you that your manuscript has been deemed suitable for publication in PLOS ONE. Congratulations! Your manuscript is now with our production department. 

Kind regards, 

on behalf of

Dr. Aleksandra Barac 

Academic Editor

PLOS ONE